# Vegetated Buffer Zone Restoration Planning in Small Urban Watersheds

Yucong Duan [1,2,3], Jie Tang [1,2,3], Zhaoyang Li [1,2,3,*], Bo Yang [4], Yu Yan [5] and Yao Yang [1,2,3]

[1] Key Laboratory of Groundwater Resources and Environment, Ministry of Education, Jilin University, Changchun 130012, China; duanyc@mails.jlu.edu.cn (Y.D.); tangjie@jlu.edu.cn (J.T.); yangyao18@mails.jlu.edu.cn (Y.Y.)
[2] Key Laboratory of Water Resources and Water Environment, Jilin University, Changchun 130012, China
[3] College of New Energy and Environment, Jilin University, Changchun 130012, China
[4] College of Agronomy, Jilin Agricultural Science and Technology University, Jilin 132101, China; ybo917@126.com
[5] China Northeast Municipal Engineering Design & Research Institute Co., LTD., Changchun 130021, China; 18686670076@163.com
[*] Correspondence: zhaoyang@jlu.edu.cn

**Abstract:** Vegetated buffer zones (VBZ) are accepted worldwide as a low impact method to avoid non-point source pollution and restore the balance of river ecosystems. Strongly influenced by industrialization and urbanization, urban river ecology is seriously damaged, and restoration is tricky. This study established a complete buffer zone construction framework suitable for the small urban watershed, and its feasibility is verified in a small watershed in Northern China. First, common plants in the study area were selected to test their ability to purify pollutants, and plant combinations were optimized. Secondly, according to the field investigation, the reference buffer zone was determined, and its sewage interception capacity was tested through a runoff simulation experiment. Then, based on GIS and Phillips time and hydraulic models, the normal buffer width of the study area was obtained; 60 m for mainstream and 40 m for tributaries. By optimizing the vegetation scheme and delimiting an efficient buffer zone, the land occupation can be reduced by 17%. Finally, combined with the characteristics of different river sections, an elaborate VBZ restoration scheme is designed from the aspects of vegetation, planning, and zoning. Generally, this research will provide government and land managers scientific and practical ideas and technologies to formulate a land management policy for urban river buffer zones in order to find a balance between aquatic ecological protection and urban land use planning and optimize the allocation of construction funds.

**Keywords:** river ecosystems; ecological restoration; plant arrangements; buffer delineation; urban planning

## 1. Introduction

Riparian buffer zones, especially vegetated buffers, are viewed as the significant barrier between aquatic and terrestrial ecosystems to protect the freshwater environment and wildlife habitat. These areas have been demonstrated to stabilize the riparian microclimate, control soil erosion, retain sediment, nutrients, and pesticides from surface runoff and provide an ecological corridor to guard biodiversity [1–5]. However, studies show that riparian buffers are threatened by anthropogenic construction activities as well as agricultural and urban landscapes due to inappropriate planning in many countries in the past decades [6–8]. Therefore, the protection and restoration of the vegetated buffer zones (VBZ) are considered a low-impact and sustainable method for the government and land users to fulfill the service value of aquatic ecosystems [9,10].

Recent findings indicate that land use in different scales may influence water quality through changing the amount and flow path of contaminants from surface runoff or

drainage network to the hydrologic systems [11,12]. Undoubtedly, accelerating urbanization brings positive changes to the economic development of suburban areas but replaces the original land cover with a more impervious surface [13,14]. However, restoration faces more challenges due to various obstacles such as land planning, sewage discharge, economic cost, and landscape demands. Thus, a buffer plan utilizing less space but with more interception capability has become a priority [15,16]. Moreover, the urban riparian landscape needs to become a green area with a high appreciation and enjoyment for lifestyle attraction.

Factors affecting the practice of buffer zones mainly focus on four aspects: the width, structure, species composition, and the vegetation management [17]. Buffer width is one of the most critical and fundamental aspects during design [18]. Typically, widths of 5–300 m are required to filter 10–95% sediments and nutrients [19–21], and scholars offer advised width through investigations, experience, or mathematical models [22,23]. Complicated models like REMM (Riparian Ecosystem Management Model) [24,25], CREAMS (Chemical, Runoff and Erosion from Agricultural Management System Model) [26], and VFSMOD (Vegetative Filter Strips Model) [27] that are based on the whole physical process of sediment and pollutant deposition and migration can simulate the minimum width for specific removal efficiency of different riparian zones [28]. Some multi-functional hydrological models like SWAT also integrate a buffer width module [29]. Methods based on geographic Information Systems (GIS) and remote sensing techniques are also highlighted for the privilege in catchment scales and can delimit variable buffer zones for multiple needs [30]. Xiang (1996) combined a math model derived from Phillips and GIS raster calculation model to identify boundary cells [31–33]. Julio Novoa et al. (2018) used an RSQI index assessed by information extracted from high-resolution images to evaluate the ecological condition of buffer zones [34]. Chun-hua Li et al. (2019) mapped the buffer zones of Zhushan Bay, Tai Lake through identifying critical source areas of non-point source pollution and ecologically sensitive areas based on GIS [35]. Furthermore, current satellite-derived images with multiple spatial and temporal resolutions and better accuracy offer studies to fit different research scales from huge river catchment scales to small agricultural plots or urban areas.

The structure of the vegetation composition strongly influences the capability of the buffer to reduce non-point sources (NPS) and create a natural environment. Typically, the structure is decided by distinct natural zones according to three-dimensional spatial structure, longitudinal, transverse, and vertical [17]. However, in urban areas, artificial interferences such as hard revetment and flood control walls lead to the separation of river and land and loss of the structural characteristics, so detailed and natural-like zoning planning and plant community structure design is required [36].

It is widely illustrated that aged woodland is the favored vegetation structure to maintain nutrient uptake, reinforce riverbanks, and stabilize underground flow for the dense underground root network formed by shallow and fibrous roots [37]. Studies have shown that the nutrient absorption of woodland plants was significantly higher than that of herbaceous plants [38]. However, the presence of vigorous herbaceous plants is indispensable, as the above-ground parts are very effective at retaining particulate matter and disperse concentrated flow during storms. Therefore, the structure with arbor, shrub and herb is recommended to create habitat diversity and sustain the ecological system [39,40]. Most management guidelines recommend it is better to select a natural riparian buffer as the reference standard and design the community structure regarding the native riparian forest to increase the species richness [41]. Native species are precious when choosing species composition, and other functional plants chosen based on climate, soil characteristics, pollution interception ability, and decorative effect must be introduced properly to prevent invasion of alien species.

The Chinese government launched the Action Plan for Prevention and Control of Water Pollution in 2015, requesting that black and odorous water bodies in urban built-up areas be generally eliminated by 2030. The Dongxinkai (DX) River was heavily polluted,

and the water quality was reduced to the worst class according to China's surface water environmental quality standards, with average chemical oxygen demand (COD), total nitrogen (TN), and total phosphorus (TP) concentrations of 309.15, 63.73, and 3.17 mg/L in 2015. Therefore, the Changchun municipal government launched the comprehensive treatment plan of black and odorous water bodies and completed the construction of a sewage treatment plant and drainage pipe network. An appropriate design of riparian buffer strips is crucial for the successful restoration of river ecology.

Overall, the purpose of this study is to develop a general idea of ecological restoration planning of heavily polluted urban rivers. For this purpose, after a detailed field investigation, we conducted experiments to choose appropriate plant arrangements and built a reference buffer to obtain the required parameters. Afterward, the VBZ width suitable for VBZ restoration and urban planning was delimited based on GIS. Finally, these conclusions and methods were applied to the planning of the Dongxinkai River buffer zone. This research may help the government manage the VBZ better to provide sustainable protection for river ecosystems.

## 2. Materials and Methods

### 2.1. Study Area

The DX River (Figure 1), located in the suburban area of Changchun, Jilin Province, northern China, runs 16.5 km and flows into the Yitong River, with 6 short tributaries about 1.9–3.5 km. The river has long been drained by summer rainfall and unpurified urban sewage, therefore producing a small average annual runoff and low water self-clarification capacity. The catchment covers about 100.9 km$^2$, defined according to administrative divisions and regional topography. The landscape is predominately covered by agriculture land (30%) and urbanized land (60%). Slopes of the catchment range from 0° in agricultural and urban areas to 35–40° along part of the riverbank. The climate is subtemperate continental monsoon, with annual temperature and rainfall averaging 4.8 °C and 571 mm. During the summer rainstorm and winter snowmelt, the risk of soil erosion increases due to the rapid surface runoff and inadequate drainage collecting network. Therefore, it is extraordinarily necessary to build a healthy and active riverbank buffer zone considering desirable buffer zone width, plant allocation, and long-term management.

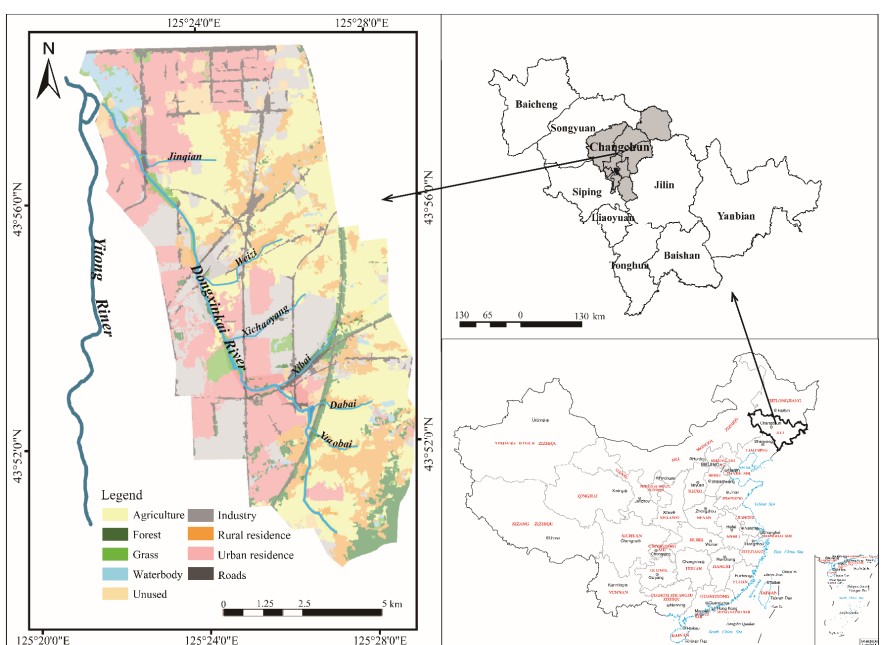

**Figure 1.** The location of DX River basin. The map of China was downloaded from http://bzdt.ch. mnr.gov.cn/ (5 September 2021).

## 2.2. Field Investigation

Field investigation was planned to receive the original data of topography, VBZ width and collect native plant types for the identification of restoration plant materials. From September to October 2017, a fieldwork survey was carried out along the main stream and all tributaries, assisted by UAV aerial photography, and more than 110 aerial photos were obtained. Furthermore, the vegetation type, land use, and topographic information of 23 plots were meticulously investigated as validations of slope calculation and land cover classification based on GIS. The width of VBZ varies from 5 m to 70 m, which is narrow in urban areas typically since the buffer zone is occupied for urban construction land and agricultural land (Figure 2). The riverbank slope is generally gentle, but there is a height difference between the buffer and the river channel in 38% of the riverbanks, which leads to severe erosion. Original vegetation types include typical northern trees and wild weeds such as green bristlegrass (*Setaria viridis* (L.) Beauv.) and humulus scandens (*Humulus scandens* (Lour.) Merr.). Moreover, according to the results of the field investigation of the 23 plots and future urban landscape construction, a 450 m demonstration plot vegetated by various kinds of herbs was constructed to be the reference buffer zone, and all the field experiments and required parameters were also obtained there.

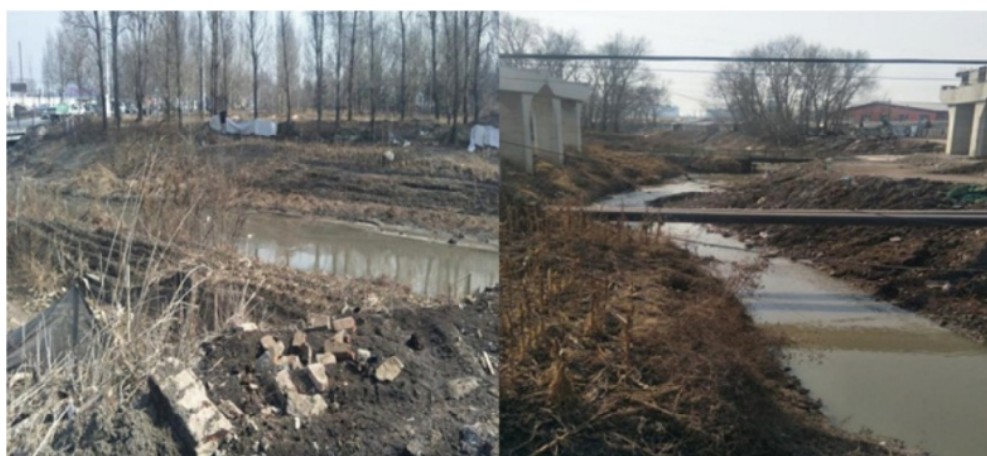

**Figure 2.** Typical VBZ along DX River before restoration. The photos were taken on 20 October 2017, before the DX river treatment project.

## 2.3. Plant Arrangement Selection

For higher biomass, biodiversity, and resistance to pests and diseases, it is best to use multiple species in a planting. The common plants in the north were selected as the experimental objects, and the plant arrangements were formed by the random combination of the plants with better pollution interception ability through pot experiments (3 cm in diameter and 28 cm in height). Experimental tanks (1.1 m × 1.0 m × 0.5 m) with a water distributor and 6 piezometers (PVC pipes; length 1 m, diameter 0.02 m) connected with sampling external tap were designed to test the detention capacity of different plant arrangements for pollutants in runoff. Nine plant materials were selected for random combination planting in experimental tanks, each a combination of four to six plants, including arbor, shrub and herb plants (Table 1). The 2-year-old seedlings with similar volume were selected, the root systems were trimmed to the same size, and then they were moved into the tanks in the middle of April and the herbaceous plants were sown in May.

**Table 1.** Plant materials of the 6 test groups. A checkmark means the plant is included. The series A, S, and H represent arbor, shrub, and herb materials, separately.

|  | Test Groups | I | II | III | IV | V | VI |
|---|---|---|---|---|---|---|---|
| A1 | *Salix matsudana* Koidz. | √ | √ | √ |  |  |  |
| A2 | *Prunus padus* L. |  | √ |  | √ | √ | √ |
| A3 | *Fraxinus mandschurica* Rupr. | √ |  |  | √ | √ |  |
| S1 | *Amorpha fruticosa* Linn. | √ |  | √ | √ | √ |  |
| S2 | *Spiraea japonica* Gold Mound. | √ | √ |  | √ | √ | √ |
| S3 | *Swida alba* Opiz. |  | √ |  |  |  | √ |
| H1 | *Medicago sativa* Linn. | √ |  |  | √ |  | √ |
| H2 | *Trifolium repens* Linn. | √ | √ | √ |  | √ | √ |
| H3 | *Festuca arundinacea* |  |  | √ | √ |  |  |

The synthetic wastewater was prepared using $KHCO_3$, $KH_2PO_4$, $CH_3COONa$, $NH_4Cl$, $C_6H_{12}O_6$, etc., and was slowly over-irrigated into the tank. The operation was repeated every 7 days for 5 periods, and the blank test of no plant and clean water irrigation was set at the same time. All inlet and outlet samples were analyzed 3 times for TP, TN, and COD using standard spectrophotometry analytical methods. The removal efficiency R was calculated by:

$$R = \frac{c_{in} - c_{out}}{c_{in}} \times 100, \tag{1}$$

where $c_{in}$ (mg/cm$^3$) and $c_{out}$ (mg/cm$^3$) are concentrations of inlet and outlet samples, respectively. All statistical analyses were performed using SPSS17.0 for Windows (SPSS, IBM, Armonk, NY, USA)

*2.4. VBZ Delineation*

2.4.1. Reference Buffer Zone

In order to facilitate the landscape planning and the ecological restoration of the riparian zone, a standard should be established. Scholars proposed the concept of reference systems to define and describe the real state of the riparian zone under natural conditions [42–44]. However, rivers flowing through the cities are likely to be regarded as artificial landscapes to provide other functions as well compared to other rivers. Therefore, the artificial grassland vegetated by different types of herbs in the demonstration plot was identified as the reference in this study instead of natural buffer, and the basic parameters and interception efficiency of runoff pollutants were observed. The study plots (10 m × 5 m) were built for the pollutants retention experiment, two of them were planted with common herbs in the north (P1, P2), such as alfalfa, etc., and the other one was treated as a blank control with no plants (B).

Experimental design and sampling points are shown in Figure 3. During the test, the uniform water distribution flow from the top was subjected to 2.5 m$^3$/h, which constituted of the rainfall intensity of a rainstorm, and continued for 60 min. The inlet and outlet water samples were collected every 20 and 10 min, separately. The collected water samples were stored in 1000 mL plastic bottles and taken back to the laboratory for testing for TP, TN, and ammonium ($NH4^+$) within 24 h. The unfinished water samples were refrigerated at 2–5 °C. The efficiency R was calculated by Equation (1) and regarded as the removal capability of reference buffer on these pollutants.

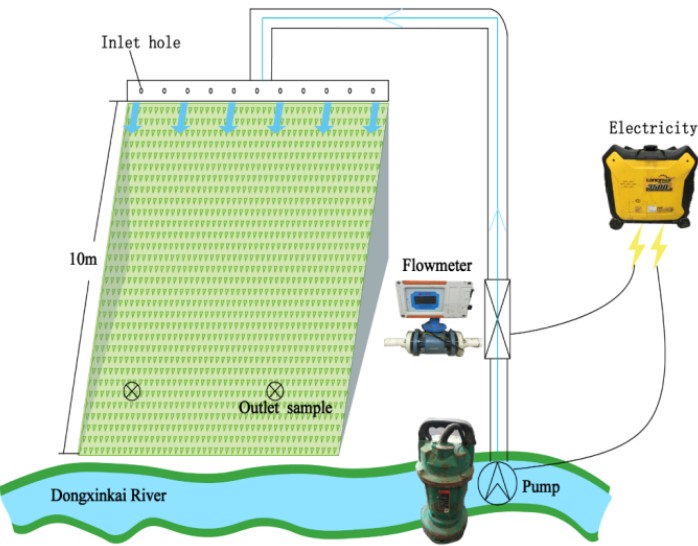

**Figure 3.** Field experiment design diagram.

2.4.2. Data Source

The database integrates four groups of spatial data: (1) digital elevation model (DEM) of the planning region; (2) raster data on hydrographic networks; (3) vector data on land use; and (4) vector data on soil type. In order to ensure the urban landscape and economic development, the natural space of urban rivers is limited, so an appropriate scale and accuracy are important for the planning of VBZ in different river basins. Accordingly, the common commercial DEM data and remote sensing images are not available for the spatial resolution and vertical accuracy of VBZ planning of small urban rivers. Thus, elevation point cloud was used to model DEM and develop the slope layer. The land cover map was based on the remote sensing image of Sentinel-2, with 10 m resolution, to divide the land use into 9 types through the sample-based object-oriented classification module of ENVI 5.3 (Figure 4). Moreover, the field survey ensured the calibration and validation of the classification. Soil type distribution was obtained from Harmonized World Soil Database (HWSD) of FAO (https://www.fao.org/soils-portal/soil-survey/soil-maps-and-databases/harmonized-world-soil-database-v12/en/, 3 April 2021). All spatial data were stored and processed by ArcGIS 10.2 software (ESRI, Redlands, CA, USA). The data sources are summarized in Table 2.

**Table 2.** Data sources and a detailed description.

| Data Type | Data Source | Data Usage | Type |
|---|---|---|---|
| Hydrology | Changchun river map; field survey | River network raster image | Grid 2.5 m × 2.5 m |
| DEM | Elevation point cloud | Slope raster | Grid 2.5 m × 2.5 m |
| Land use | Sentinel-2 multispectral image (https://scihub.copernicus.eu/, 3 April 2020); field survey | Manning coefficient raster | Grid 10 m × 10 m |
| Soil type | UN Food and Agriculture Organization (FAO) | Saturated hydraulic conductivity raster and soil moisture storage capacity raster | Grid 1 km × 1 km |
| Reference buffer parameters | Field survey | Basic parameters | ASCII text |
| UAV image | UAV shooting | Assist field survey | JPG |

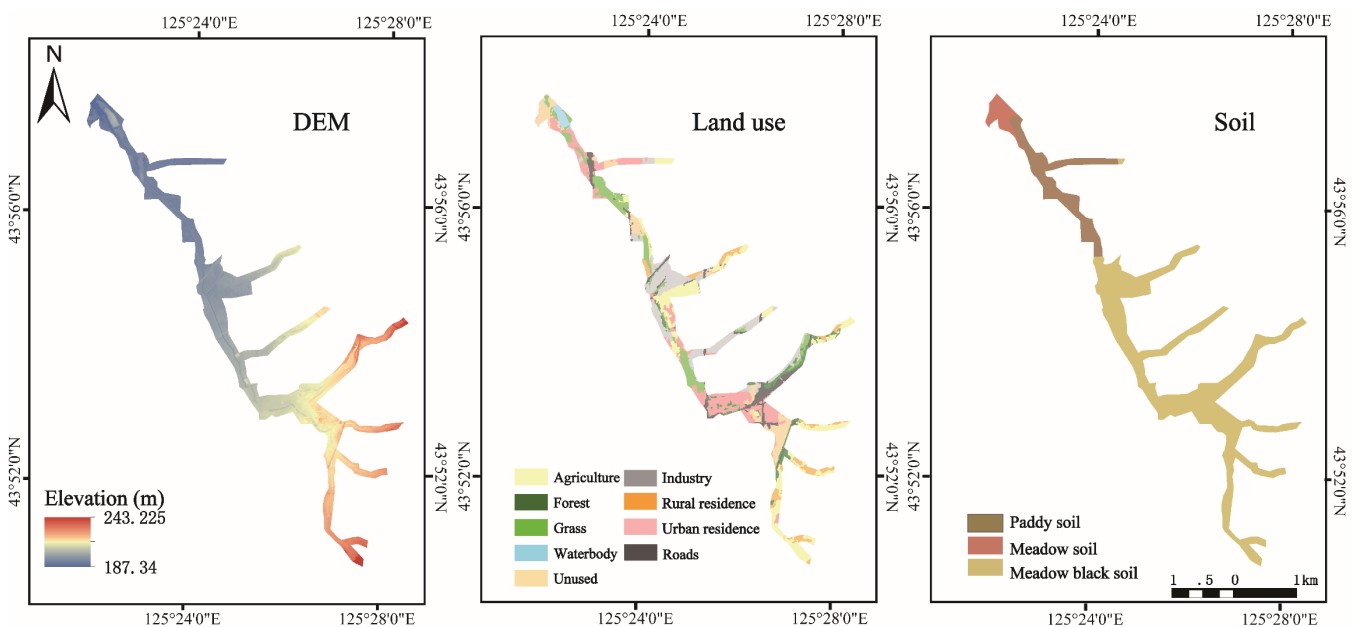

**Figure 4.** DEM, land use, and soil type map of DX river construction area.

### 2.4.3. Determination of VBZ Boundaries

The advantage of VBZ for river water quality protection is its capability to remove and retain non-point source pollutants in surface runoff, which depends on its potential to absorb, retain and even decompose pollutants. Rainfall-overland flow through VBZ is expected to be even, turbulent and slow, which may create a longer detention time before entering the river. These conditions may be likely to obtain a high-quality VBZ with low erosion and high interception capacity. This potential is affected by topography, soil properties, land cover, and other factors, and the removal effect of dissolved and floating pollutants is different. Phillips (1989) proposed a time model and hydrological model to calculate the width of riparian VBZ, a model based on physical relationships and assumptions that are standard in hydrologic analyses [31]. The model is spatialized based on GIS and was applied in some watersheds, such as North Carolina and Shanghai [31,45,46]. In this study, we applied the model in DX river basin to map a desirable fixed and variable buffer width.

1. Model of buffer width

The time model is a mathematical model for removing dissolved pollutants, including the absorption and transformation in the VBZ. The pollutants' retention time can be available to represent the efficiency of riparian vegetation buffer:

$$T_b/T_r = B_b/B_r = (n_b/n_r)^{0.6} (L_b/L_r)^2 (K_b/K_r)^{0.4} (s_b/s_r)^{-0.7}(c_b/c_r), \qquad (2)$$

where b: proposed buffer, r: reference buffer, T: detention time of surface flow, $B_b/B_r$: buffer effectiveness ratio, n: Manning roughness coefficient, L: buffer width (m), K: saturated hydraulic conductivity (cm h$^{-1}$), s: slope (%), and c: soil moisture storage capacity (cm). To simplify, p represents the buffer effectiveness ratio:

$$p = B_b/B_r, \qquad (3)$$

combining Equation (3) to Equation (2), it becomes:

$$L_p = p^{0.5}L_r [(n_r/n_b)^{0.6} (K_r/K_b)^{0.4} (s_r/s_b)^{-0.7}(c_r/c_b)]^{0.5}, \qquad (4)$$

where $L_p$, is the desired width to reach a buffer effectiveness ratio p, compared with the reference buffer. The basic parameters of the reference buffer which were collected in the field survey are listed in Table 3.

**Table 3.** Parameters of the reference buffer.

| Reference Buffer Index | Value |
| --- | --- |
| Width ($L_r$) | 10 m |
| Slope ($s_r$) | 18% |
| Soil moisture storage capacity ($c_r$) | 32.9 cm |
| Saturated hydraulic conductivity($K_r$) | 0.71 cm h$^{-1}$ |
| Manning roughness coefficient ($n_r$) | 0.58 |

The hydraulic model is a mathematical model for intercepting suspended solid pollutants from the surface runoff. Only the energy loss in hydraulic process is considered, thus, the model is defined by three factors: the width, the slope and the surface roughness [47]. Therefore, the hydraulic model is given by:

$$P_b/P_r = B_b/B_r = (n_b/n_r)^{0.6}(L_b/L_r)^{0.4}(K_b/K_r)(s_b/s_r)^{-1.3}, \tag{5}$$

Same as above, adding Equation (3) to Equation (5), it becomes:

$$L_p^* = p^{2.5}L_r\left[(n_r/n_b)^{0.6}(K_r/K_b)(s_r/s_b)^{-1.3}\right]^{2.5}, \tag{6}$$

where $L_p^*$ is the desirable width for intercepting solid pollutants.

If the buffer effectiveness ratio p equals 1, an appropriate buffer width can be calculated for the design buffer to reach the same interception potential as the reference buffer. Manning roughness coefficient n is a measure of the plant arrangement in this model. Therefore, two scenarios based on different values of $n_b$ were designed for midstream and downstream flow through the urban area and urban-rural area, respectively.

2.    VBZ mapping based on GIS

To map a variable VBZ, a GIS database was first developed that contains four spatial layers, c, K, n, and s, which are needed for width calculation. Second, a combined data layer was created by a GIS overlay function to calculate the minimum width $B_{wi}$ for different landscape patches when the buffer effectiveness ratio reaches p. Then, parameter $c_i$ was introduced as the unit contribution index of pollutant removal:

$$c_i = \frac{1}{B_{wi}}, \tag{7}$$

and the layer was converted from a vector to raster format, with 2 m resolution. In this research, we concentrate on the accumulated $c_i$ from each cell to the river, namely regarding the $c_i$ raster as cost raster and the hydrology as source raster to calculate the travel cost from each of the river buffer cells to the nearest source location. We automated the process with the cost distance tool of ArcGIS10.2 (ESRI, America) to get a grid file, each grid value represents the least accumulative cost distance of the corresponding 2 m × 2 m cell over a cost surface to the identified river source locations (the specific operation process can be referred to: https://desktop.arcgis.com/en/arcmap/10.4/tools/spatial-analyst-toolbox/cost-distance.htm, 4 September 2020). Finally, the cells with a value of *p* are identified as buffer boundary cells in the output raster and converted the buffer to a polygon file. The defined buffer based on the time and hydraulic model were overlaid to be the final variable river buffer. Furthermore, an averaged fixed width calculated by the total area of VBZ divided by river length was considered as a recommended VBZ width for managers.

## 3. Results

### 3.1. Plant Arrangements Optimization

During the experiment, the growth of the plants was in good condition, without death or diseases, which suggests that the test plants are tolerant to the sewage to some extent. In plant growth, pollutants in sewage water are constantly absorbed and utilized, providing certain nutrients for plants. The purification rate of each plant arrangement for different pollutants is shown in Figure 5. It was observed that there is a significant difference between the removal efficiency *R* of the 6 test groups ($p < 0.05$). The values of TN, TP and COD ranked as I > III > V > II > IV > VI, I > III > II > V > IV > VI and II > IV > III > I > VI > V. The arrangements I and III showed better purification capability on TN and TP, valued at 52.6–78.2%. However, II was the best on COD, reaching 62.4%, followed by combination IV. In general, II and III showed stronger comprehensive interception capacity. Among these arrangements, willow showed strong performance with fast growth and good pollution interception ability, consistent with previous research results [48,49]. *Amorpha fruticosa* and *Alfalfa* had advantages in the tested shrubs and herbs. To summarize, the groups with larger biomass and more nutrient needs showed a stronger ability to remove phosphorus and nitrogen.

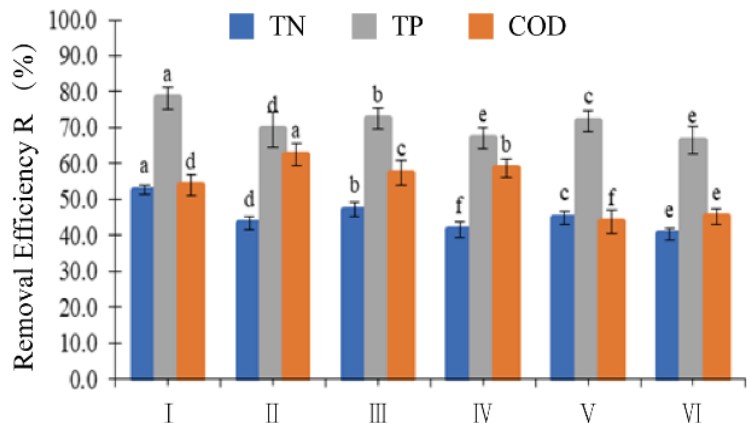

**Figure 5.** The removal efficiency R of TN, TP and COD by 6 plant material groups. The letters a–e indicate multiple significant difference results.

When a variety of plants grow together for a period of time, their mutually beneficial symbiotic relationships will reach a stable formation of community, and they will fulfill a variety of ecological benefits together. When the system is stable, it is recognized to surpass the population of a single plant in biodiversity, biomass, resistance, landscape ornamental, and other aspects.

### 3.2. VBZ Delineation and Design

#### 3.2.1. Reference Buffer

Considering that the climate and growth period will significantly affect the interception efficiency of various pollutants and nutrients in the VBZ, we chose the middle period of herbage growth to carry out the field experiment and determine the required parameters for subsequent VBZ delineation. The results showed that, during the 60 min experiment, as for N, all the plots acted as a N retaining sink, 23.5% for P1 and 24.2% for P2, while there was a clear trend of higher absolute removal of vegetated plot P1 and P2 than B (Figure 6). As for different forms of N, $NH_4^+$-N removal efficiency R in the plots valued higher than TN, due to the root absorption during warm growing seasons and the water-saturated and anoxic conditions which lead to the denitrification process in the soil [50,51]. However, as for P, the vegetated plot acted as a net P sink, but the unplanted plot B turned out to be a net source of P throughout the simulation experiment, and lower

interception ability of dissolved total phosphorus (DTP) was observed. This result can be interpreted that P usually exists in the particulate form on the soil surface, so plant stems and roots will play an essential role in intercepting soil particles during rapid runoff. In addition, redox-sensitive iron (Fe)-P combined reduction solutions may lead to increased phosphorus release [52].

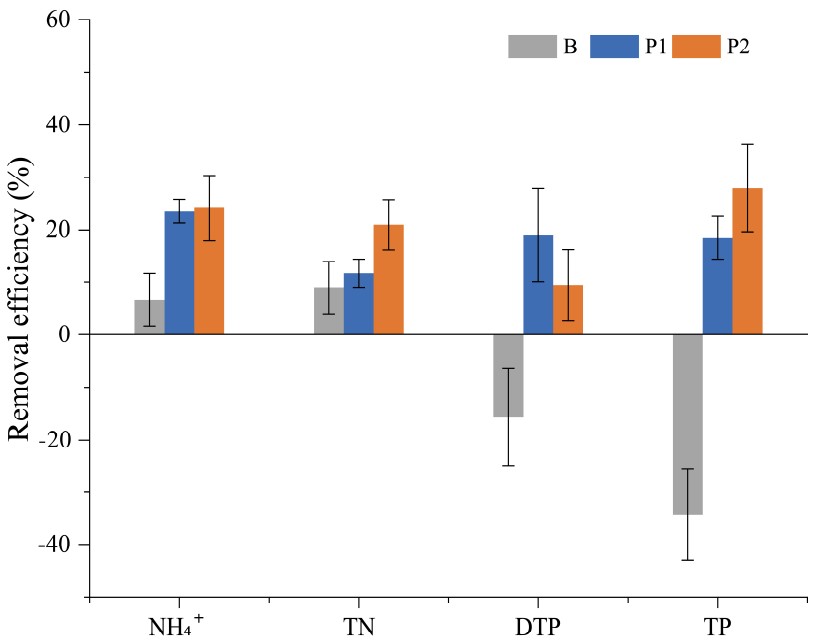

**Figure 6.** Average removal efficiency of ammonium ($NH_4^+$), total nitrogen (TN), dissolved total phosphorus (DTP), and total phosphorus (TP) of three plots in 60 min.

To summarize, in the study area, the pure herb VBZ with a slope of about 15 degrees and in the middle stage of growth can intercept at least 20% of the storm runoff generated on the slope, which can be considered as the lowest standard for later designs and utilized as reference buffer.

### 3.2.2. Normal Buffer Width Identification

The width of ideal VBZ defined by time model and hydraulic model for different patches varies significantly, which is obviously affected by slope and land cover conditions. Patches with a steep slope and low roughness show higher demand for buffer width. There are some differences between the boundary delimited by the time model and hydraulic model; the time model boundary is stable and uniform, ranging from 5 m to 20 m. In contrast, the hydraulic model boundary is more variable with a minimum of 7.9 m to a maximum of 176 m (Figure 7). The sensitivity of slope parameters is noticeable, a tiny increase of slope value will greatly improve the width of model operation, which is consistent with some studies that considered microtopography a crucial factor to affect runoff occurrence and transport and pollutant retention and absorption [47].

In order to facilitate the management and planning of the river basin by government departments and landowners, after mapping the variable buffer, the minimum area and the recommended width of the VBZ that should be guaranteed for the mainstream and each tributary were summarized (Table 4). The mainstream of the river is surrounded by urban construction land with a steeper riverbank, so its recommended the width becomes the widest, more than 6 m. For all tributaries, the recommended width of Xiaobai and Xibai are wider, covered 41.03 m and 37.26m, since Xiaobai is located in a hilly area with high terrain and steep slope and both sides of Xibai are covered by industrial and transportation land. In general, the 40–60 m wide buffer zone can guarantee the interception of 20–25% nutrients in the rainfall runoff, which can meet the demand of non-point source pollution

control in the basin, thus improving the water quality and reducing the sudden input of pollutants.

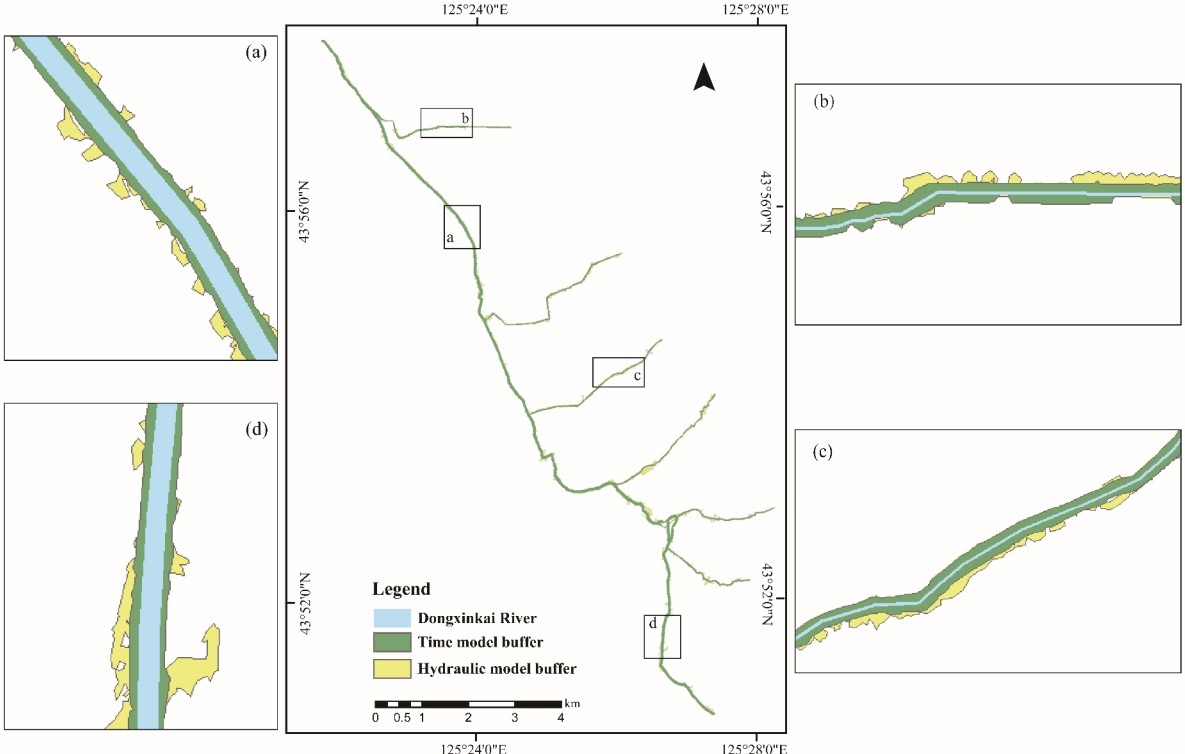

**Figure 7.** Delimited VBZ boundaries for time model and hydraulic model. (**a**–**d**) are zoomed in on specific areas.

**Table 4.** Normal VBZ width of DX River.

| River | Time Model | | Hydraulic Model | | Union |
|---|---|---|---|---|---|
| | Area (m$^2$) | Width (m) | Area (m$^2$) | Width (m) | Width (m) |
| Mainstream | 927,528 | 55.21 | 914,256 | 54.42 | 63.19 |
| Xiaobai | 72,409 | 38.11 | 56,202 | 29.58 | 41.03 |
| Xibai | 73,416 | 26.22 | 69,580 | 24.85 | 37.26 |
| Weizi | 97,755 | 27.93 | 89,355 | 25.53 | 34.18 |
| Dabai | 63,986 | 27.82 | 51,106 | 22.22 | 33.41 |
| Xichaoyang | 65,400 | 21.80 | 81,450 | 27.15 | 32.00 |
| Jinqian | 74,675 | 25.75 | 69,687 | 24.03 | 31.41 |

### 3.2.3. High Efficiency Buffer Width Identification

Based on previous research, we tried to represent the optimization of the plant cover of the VBZ by changing the value of the Manning coefficient to simulate the high efficiency VBZ that may have a higher interception efficiency than the reference buffer zone, to reduce some buffer width to alleviate the contradiction of land use shortage in urban small watersheds. In addition, the width and design idea of the high efficiency VBZ are applied to the river section located in the urban area, including the midstream and the tributary of Xibai as the heavily polluted urban river section, and downstream as the urban-rural river section. According to the simulation results of the model, after vegetation optimization, the land occupation area of the VBZ can be slightly reduced. The integration width of the VBZ in the middle and lower reaches of the mainstream can be reduced from 63.19 m to 48.06 m and 47.02 m, and the width of Xibai can be dropped from 37.26 m to 21.96 m (Table 5). This adjustment applied in 46% of river length can save about 17% of land compared with the normal buffer for the economic and social development of the basin. It can be

considered that optimizing vegetation allocation and planting technology can fulfill the balance between urban land and river ecological protection to some extent. However, the premise is that regular maintenance and long-term supervision must be guaranteed, so the cost of economy, workforce, and time must be higher. Therefore, the restoration plan can be adjusted reasonably by measuring the economic and cost value.

**Table 5.** High efficiency VBZ width of DX River section in urban areas.

| Section | Length (km) | Normal Width (m) | High Efficiency Width (m) | Section Type |
|---|---|---|---|---|
| Midstream | 6.6 | 55.21 | 48.06 | Heavy polluted urban reaches |
| Downstream | 6.0 | 55.21 | 47.02 | Urban-rural reaches |
| Xibai | 2.8 | 37.26 | 21.96 | Heavy polluted urban tributaries |

### 3.2.4. VBZ Design

A total of 37 species of plants, including 16 arbors, 13 shrubs, 6 terrestrial herbs, and 2 hygrophyte plants were selected in the whole design of VBZ, and these plants were composed of trees, small trees, shrubs, patterns and herbs, with five layers of three-dimensional structure and distinct levels. Considering the pollution interception effect of plant materials, no fruit tree species were selected in this program. In arbor and shrub areas, native trees are the main ones, with abundant tree species and deep roots, which can conserve water resources, consolidate embankments and adjust the climate. Shrubs are planted in the space under the forest to absorb pollutants, and herbaceous plants are planted in the interval zone. Based on the pollution characteristics of different river sections, three sets of VBZ repair technology models were formed, considering the high efficiency VBZ width, structural zoning, selected plant arrangement, landscape effect, etc. (Table 6). In the process of efficient VBZ design, the following principles are followed as far as possible: (1) to meet the function of flood discharge and water storage in small watersheds and reduce soil and water loss; (2) by optimizing plant configuration, planting high pollution intercepting plants and appropriately reducing the land occupation of urban buffer zone; (3) achieving efficient interception and reduction of NPS on both sides of the river and absorbing some harmful gases in the air; (4) improving biomass, greening configuration with distinct levels and seasons; (5) it is convenient for maintenance and management, and can effectively prevent the occurrence of diseases and insect pests. Fortunately, these design concepts and research results were applied to the ecological restoration project of DX River as the basic support for the restoration and maintenance of VBZ.

**Table 6.** The VBZ restoration design plan of DX river.

| Zone | Component | Width | Location Examples | Schematic Diagram |
|---|---|---|---|---|
| Midstream heavy pollution reach | I: arbor-shrub | 15–20 m | | |
| | II: herb | 5–10 m |  |  |
| Downstream urban-rural reach | I: arbor-shrub | 15 m | | |
| | II: shrub-herb | 5–10 m |  |  |
| | III: herb | 5 m | | |
| Rural weak management reach | I: arbor | 15–20 m | | |
| | II: herb | 5 m |  |  |

## 4. Discussion

The design of river buffer zone in urban built-up areas is different from that in large river basins, which needs to be more detailed and scientific. Plant selection and allocation, reasonable buffer width, and sustainable management are equally important. Plant selection is the basis of green coverage, and priority should be given to local species; alien species for promoting ornamental effect must be tested before large-scale planting. Moreover, the number and proportion of different plants, planting position, and density are acquired to be designed scientifically in advance to plant the available species together and form a sustainable ecosystem. Configuration of arbor, shrub, and herbage varies due to different construction purposes; the arbor has some advantages in stabilizing riverbanks, protecting groundwater, and resisting flood; the grassland is well-suited to filtering nutrients and insecticides, and improving animal habitat.

Buffer zone boundary delimitation is the main idea to preserving the ecological space for river ecosystems. Delineation methods based on GIS technology are feasible and reliable which can produce meticulous results through spatial calculation affected by topography, soil, and surrounding conditions. The accuracy of the boundary mapping highly depends on the spatial resolution of the acquired land cover and slope data, so it is essential to choose a suitable accuracy level to fulfill the basin size, planning purpose, and project budget. For elevation data set, LiDAR technology and airborne LiDAR system are gradually common in topographic information collection and positioning repeated observation, which can quickly acquire large area 3D terrain data and produce digital products [53]. Research in ecology has laid more emphasis on the unmanned aerial vehicle (UAV) technology through the whole process of design, construction, and supervision in recent years because of its advantages in vegetation survey, routine monitoring, and emergency investigation [54]. In this research, we applied the technologies appropriately to reach the scale and accuracy we require without bearing a great economic burden. Undoubtedly, the extensive application of these technologies creates new vitality for some classical and precise models.

In the future, almost all population growth will concentrate in urban areas and bring more pressure on urban drainage and river landscape design, especially in developing countries. Most countries started the history of river protection from the treatment of water pollution, then to the reconstruction of small river habitats, and finally to the restoration of the whole river ecology due to the lack of long-term land use planning, and the urban ecosystem was destroyed repeatedly. In order to protect the economic development and reserve a specific living space for the aquatic ecosystem, it is necessary to draw a reasonable control boundary for the river in advance, prohibit destructive human activities within the boundary, and carry out vegetation restoration to conserve water resources and protect the habitats of animals and plants. Urban rivers can never be regarded as a part of the linear drainage pipeline, but a strip-shaped reserved land, which should first be circled in the urban land use planning and guaranteed by the land use policy. Furthermore, the focus of aquatic ecological restoration should gradually turn to the protection of aquatic plants and their habitats by optimizing the water quality evaluation system and implementing a typical water ecological function zone pilot. It is commonly recognized that beautiful urban water spaces and landscapes are more and more critical while disposing the city resources to promote the city vitality and competition capability. This study was developed in Changchun to provide a recommendation process for the local government to build scientific buffer zones, and ensure that the environmental protection, ecological value, and operating costs are effectively taken into account in the restoration and maintenance of riparian zones. Its design ideas could be used for references and applied to other urban watersheds in cities and suburbs.

## 5. Conclusions

This research proposed general ideas and technologies for the planning of riparian buffer zone restoration in heavily polluted urban watersheds and applied the whole process in the DX River, Changchun, China. It forms a complete integrated buffer planning and

design process, including width delimitation, vegetation screening, zoning design, and field verification. In the study area, the 40–60 m VBZ can intercept 2025% of nutrients from the storm runoff. The order of nutrient interception capacity of VBZ was $NH_4^+$-N > TN > TP. By optimizing vegetation structure, the width of the buffer zone can be reduced to around 40 m and 20 m, which saves 17% of the available land for future economic development. According to the characteristics of different river sections in the basin, three configurations of the arbor, shrubbery, and grass in the buffer zone were designed with an emphasis on pollution interception, landscape, and cost, separately. In addition, this study puts forward the design principle of the buffer zone in the small urban watershed, which can alleviate the contradiction between urban land shortage and river ecological protection, and provide suggestions and references for other similar research areas.

**Author Contributions:** Y.D.: Conceptualization, Methodology, Writing—Original Draft. J.T.: Resources, Supervision. Z.L.: Visualization, Investigation. B.Y.: Experiment, Data Curation. Y.Y. (Yu Yan): Data Curation, Reviewing. Y.Y. (Yao Yang): Writing—Reviewing and Editing. All authors have read and agreed to the published version of the manuscript.

**Funding:** This work was supported by the Department of Science and Technology of Jilin Province, Key scientific and technological projects, "Study on Ecological Protection and Restoration Technology of Buffer Zone of Dongxinkai River, a Heavy Polluted Tributary of Yitong river" (20170204001SF) and the Major Science and Technology Project, Science and Technology Department of Jilin Province "Pollution Prevention and Control and Ecological Restoration in Liaohe River Basin of Jilin Province" (20200503003SF).

**Institutional Review Board Statement:** Not applicable for this study.

**Informed Consent Statement:** Not applicable for this study.

**Acknowledgments:** Thanks to all of the co-authors for their help in this study, and especially thanks to the Dongxinkai River black odor water treatment project team for providing basic data and research site for this study.

**Conflicts of Interest:** We declare that we have no financial and personal relationships with other people or organizations that can inappropriately influence our work, there is no professional or other personal interest of any nature or kind in any product, service and/or company that could be construed as influencing the position presented in, or the review of, the manuscript entitled, "Vegetated Buffer Zone Restoration Planning in Urban Small Watersheds." Additionally, there is no economic contradiction between the authors.

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
