# Peer review of "Vegetated Buffer Zone Restoration Planning in Small Urban Watersheds"

_water, doi:10.3390/w13213000_

Round 1
Reviewer 1 Report
There is no problem with the structure or content, but the article is not in the level of 'water' journal. Maybe can be printed in another Q3/Q4 MDPI journal.
Sometimes I read very general content and sometimes the same explanation repeating. And I see such a long phrases that can be shorter based on their contents.
Abstract is quite short and do not satisfy the requirement of an abstract.
row 73: what is NPS? Explanation of abbreviation
row 89. Based on bibliography the citation 41 and 42 are the same. After delete of 42, you must recalculate the numbers of citated article.
row 97. COD, TN, TP abbreviations are here, but you explain them only later at rows 163-164.
Why do not you put space between the number and its dimension? (see fi. rows, 116, 118, 134 etc.)
Title of fig.1. is another page than the figure itself.
row 130-132: Number of UAV images? Approximatelly?
row 132: How large is a pot?
row 138. Latin names of vegetation?
Fig.1. Space between images and above the image as well.
Table 1. and Table 2: too much space between lines.
row. 177: "...to provide other functions as well than other rivers."
Ch. 2.4.2. I can not see the difference between scales, spatial resolutions.
Point density of Lidar measurement? FAO map scale?
Table 2. UAV images were used? Thes are missed here.
row 220: Number of this citation? Maybe 31, but there is nothing at the end of this phrase.
How did you calculate the weight at eq.2. and eq.5? You mean the slope is the most weighted element in the eq?
row 252: I mean big 'C' should be little 'c' here?
row 275-276: 'citation of 'figure 5' is enough one time.
row 277-278: That means II and III ae in front of the queues at different cases?
Figure 6: Why is it black and white if the others are not?
Discussion: row 405: climate data mentioned, but there were any climate data in the analysis.
citation 55 is good to here? Is beach dune system a good example?
References:
6 and 24: hy are big letters?
"et al." is not a good form in the biography? - eg. 8, 10, 12, 13, 20, 21, 23, 24, etc.
8. Typing problem in the text.
31. Philips, J.D. just one time
41 and 42 are the same.
Reviewer 2 Report
In this paper, the authors review varying widths of vegetated buffer zones and their effectiveness of filtering out pollutants in urban watersheds in China. While it is well known that vegetated riparian areas are crucial for filtering of runoff, the authors here present it in a way that helps landscape managers determine what is the proper width for site-specific conditions. I think that this paper, once polished a bit, will be fine for inclusion in Water.
First of all, I suggest a thorough grammar review, I have marked many instances in my comments below, but there are probably others that I have overlooked.
Line 4, “1” is not in superscript on last author
Line 13: make zone plural (zones) and change “is” to “are”
Line 14: make ecosystem plural (ecosystems)
Line 16: add “a” before “built-up”
Line 17: make experiment plural (experiments)
Line 19: change image to “imagery”
Line 28: Typically, you don’t reuse word from the title as a keyword, since the title is already searchable. I would suggest coming up with some new keywords
Line 32: put a comma after “vegetated buffers”
Line 36: place an “an” between “provide” and “ecological”
Line 37: change “even occupied by” to “as well as”
Line 39: change zone to “zones”
Line 41: change ecosystem to “ecosystems”
Line 44” remove “the” from before “accelerating”
Ling 45: change bring to “brings”
Line 46: change replace to “replaces”
Line 49: change to “buffer plan utilizing less space but with more interception capability becomes a priority”
Line 52: I suggest adding a colon “:” after aspects
Line 53 add a comma after “composition”
Line 57 add “that are” before “based on the whole”
Line 60: add “a” before “buffer width module”
Line 63: I suggest changing the second “variable” to “multiple” because you used variable twice real close together
Line 63: also, change “zone” to “zones”
Line 66: change “zone” to “zones”, and change “map” to “mapped”
Line 67: change “identify” to “identifying”
Line 71: change agriculture to “agricultural”
Line 84: change “retain” to “retaining” and “particular” to “particulate”
Line 85: change “recognized” to “recommended”
Line 94: change requested to “requesting”
Line 96: I suggest adding a brief discussion on the water quality classifications. You have that the site is a class V, but what does that mean in relation to the other classes?
Line 97: This is the first mention of COD, TN, and TP, they need to be spelled out on the first use, with the abbreviations in parentheses, e.g. “total nitrogen (TN)”
Line 99: change water body to “water bodies
Line 100: insert “a” before “sewage treatment plant”
Line 101: I suggest changing “following” to “successful”
Line 102: Suggest changing to “Overall, the purpose of this study is to develop a general idea”
Line 103: change “conduct” to “conducted”
Line 109: change ecosystem to “ecosystems”
Line 118: can you explain what is meant by “construction land?” in my mind, construction is the process of building something, eg a construction site. Do you mean urbanized land here? I would suggest rewording or defining the term.
Line 117: change to “The landscape is predominately covered by agricultural land”
Line 125: First off, the figure caption should be on the same page as the figure. Second, the labels in the overview map of China in the lower right is not legible. Perhaps make it larger, or remove it. Finally, the small inset map in the lowermost right corner is completely unrecognizable. I am not sure of this is even needed.
Line 132: I am going to be honest, as a native English speaker, I have never heard of “detailedly” but I looked it up and it does exist, so technically, this is write. However, it is very confusing to read/say aloud. You can leave it, but may I suggest other words such as “thorough, meticulously, comprehensively, systematically, etc”
Line 135: change agriculture to “agricultural”
Lines 147-149: This sentence is confusing. I feel like you are trying to say that it is best to use multiple species in a planting, but I am not certain. Need to work on rewording this
Line 161: change to “The operation was repeated every 7 days”
Line 163: these abbreviations need to be spelled out on line 97 where they are first used.
Line 173: insert “the” before riparian
Line 174: change system to “systems”
Line 181: change to “for the pollutant retention experiment”
Line 186: change continue to “continued”
Line 189: no need to spell out phosphorus and total nitrogen again. Ammonium is a new term, so go ahead and spell it out
Line 197: insert “and” before (4)
Line 202: change “cloud point” to “point cloud”
Line 205: insert a space between ENVI and 5.3
Line 207: add a citation for the FAO database
Lines 209 and 210: Figure 4 is mentioned in text before table 2, so these should be swapped
Line 210: on the DEM legend, please add (m), assuming that it is in meters
Line 220: insert “a” after “proposed”
Line 221: change zone to “zones”
Line 223: change “watershed” to “watersheds”
Line 252: is “C” (uppercase C) the same as the c (lowercase c) in the previous equations. If it is the same, then it needs to be lowercase. If it is a new variable, please describe it.
Line 271: suggest rewording to “During the experiment, the growth of the plants were in good condition,”
Line 279: insert “at” before 52.6%
Line 289: reword to “period of time,”
Line 307: DTP needs to be spelled out in the main text. Right now it is only spelled out in figure 6
Line 323: change “range” to “ranging”
Line 324: I suggest either placing a semicolon after “m” or ending the sentence right there with a period and starting a new sentence with “In contrast, “
Line 349: change “watershed” to “watersheds”
Line 357: insert “the” between “with” and “normal”
Line 400: change riverbank to “riverbanks”
Line 400: I suggest changing “expert in” to “well-suited to”
Line 411: change “on ecology” to “in ecology”
Line 411: this is the first mention of UAV, need to spell it out
Lines 439-441: This is placeholder text that needs to be removed
Line 459: new development may mean that the previously identified buffer width is no longer viable, just something to keep in mind that it may be an evolving process and land use changes.
Reviewer 3 Report
The paper “Vegetated Buffer Zone Restoration Planning in Urban Small Watersheds” focuses on the restoration plan of the vegetated buffer zone from the perspectives of vegetation, planning and zoning in Dongxinkai river basin. In order to achieve this objective, high-accuracy digital elevation model (DEM), high-resolution satellite image and Phillips buffer model were integrated in GIS to complete the automatic grid delimitation process of fixed and variable width buffer zone. To mapping a variable buffer zone, the authors developed a GIS database that contains four spatial layers (soil moisture storage capacity, saturated hydraulic conductivity, manning roughness coefficient and slope), created a combined data layer in order to calculate the minimum width for different landscape patch when the buffer effectiveness ratio reaches a threshold value and computed a contribution index of pollutant removal. The results are validated on field measurements and verification. The results of pot experiments demonstrated the performance of different plant arrangements. It was found that by optimizing vegetation structure, the width of buffer zone can be reduced to around 40m and 20m.
Recommendations for authors: to insert geographical coordinates in Fig. 1, 4 and 7.
Round 2
Reviewer 1 Report
Your article was getting better with your answer and generally congratulation for your work!